# Corynebacteria of the *diphtheriae* Species Complex in Companion Animals: Clinical and Microbiological Characterization of 64 Cases from France

Kristina Museux,[a] Gabriele Arcari,[b] Guido Rodrigo,[a] Melanie Hennart,[b,e] Edgar Badell,[b,c] Julie Toubiana,[b,c,d] Sylvain Brisse[b,c]

[a]Cerba Vet, Massy, France
[b]Institut Pasteur, Université Paris Cité, Biodiversity and Epidemiology of Bacterial Pathogens, Paris, France
[c]Institut Pasteur, National Reference Center for Corynebacteria of the diphtheriae complex, Paris, France
[d]Department of General Pediatrics and Pediatric Infectious Diseases, Hôpital Necker-Enfants Malades, APHP, Université de Paris, Paris, France
[e]Collège doctoral, Sorbonne Université, Paris, France

**ABSTRACT** Corynebacteria of the *diphtheriae* species complex (CdSC) can cause diphtheria in humans and have been reported from companion animals. We aimed to describe animal infection cases caused by CdSC isolates. A total of 18,308 animals (dogs, cats, horses, and small mammals) with rhinitis, dermatitis, nonhealing wounds, and otitis were sampled in metropolitan France (August 2019 to August 2021). Data on symptoms, age, breed, and the administrative region of origin were collected. Cultured bacteria were analyzed for *tox* gene presence, production of the diphtheria toxin, and antimicrobial susceptibility and were genotyped by multilocus sequence typing. *Corynebacterium ulcerans* was identified in 51 cases, 24 of which were toxigenic. Rhinitis was the most frequent presentation (18/51). Eleven cases (6 cats, 4 dogs, and 1 rat) were monoinfections. Large-breed dogs, especially German shepherds (9 of 28 dogs; $P < 0.00001$), were overrepresented. *C. ulcerans* isolates were susceptible to all tested antibiotics. *tox*-positive *C. diphtheriae* was identified in 2 horses. Last, 11 infections cases (9 dogs and 2 cats; mostly chronic otitis and 2 sores) had *tox*-negative *C. rouxii*, a recently defined species. *C. rouxii* and *C. diphtheriae* isolates were susceptible to most antibiotics tested, and almost all of these infections were polymicrobial. Monoinfections with *C. ulcerans* point toward a primary pathogenic potential to animals. *C. ulcerans* represents an important zoonotic risk, and *C. rouxii* may represent a novel zoonotic agent. This case series provides novel clinical and microbiological data on CdSC infections and underlines the need for management of animals and their human contacts.

**IMPORTANCE** We report on the occurrence and clinical and microbiological characteristics of infections caused by members of the CdSC in companion animals. This is the first study based on the systematic analysis of a very large animal cohort (18,308 samples), which provides data on the frequency of CdSC isolates in various types of clinical samples from animals. Awareness of this zoonotic bacterial group remains low among veterinarians and veterinary laboratories, among which it is often considered commensal in animals. We suggest that in the case of CdSC detection in animals, the veterinary laboratories should be encouraged to send the samples to a reference laboratory for analysis of the presence of the *tox* gene. This work is relevant to the development of guidelines in the case of CdSC infections in animals and underlines their public health relevance given the zoonotic transmission risk.

**KEYWORDS** *Corynebacterium*, diphtheria, *C. ulcerans*, *C. diphtheriae*, *C. rouxii*, emerging zoonosis, pet animal, case series

Address correspondence to Kristina Museux, kristina.museux@cerbavet.com, or Sylvain Brisse, sylvain.brisse@pasteur.fr.

The authors declare a conflict of interest. The authors declare that 2 authors (K.M., G.R.) are employees of Cerba Vet, which performs diagnostic testing on a commercial basis.

**D**iphtheria is a potentially fatal infection in humans, caused mostly by toxigenic *Corynebacterium diphtheriae* isolates, which carry the *tox* gene coding for diphtheria toxin. This bacterial species is phylogenetically related to 5 other *Corynebacterium* species (*C. ulcerans*, *C. pseudotuberculosis*, *C. rouxii*, *C. belfantii*, and *C. silvaticum*) and together with these is grouped into the *C. diphtheriae* species complex (CdSC). *C. ulcerans* (1) can be isolated from humans and animals and is being increasingly reported (2–5). *C. pseudotuberculosis* causes caseous lymphadenitis in small ruminants and edematous skin disease in buffaloes. Although this species is considered potentially toxigenic, only isolates from buffaloes in Egypt were reported to produce diphtheria toxin, and there is no evidence for toxigenicity of recent isolates from caseous lymphadenitis (6–9). Two novel species, *C. belfantii* and *C. rouxii*, were recently described (10, 11). Isolates of these species were previously identified as *C. diphtheriae*, are mostly *tox* negative, and are of biovar Belfanti (starch and nitrate negative). Last, *C. silvaticum* was recently described from wild boars; all isolates of this species carry a disrupted *tox* gene, impairing their capacity to produce diphtheria toxin (12). Hence, although all members of the CdSC may potentially harbor the *tox* gene, toxigenic strains are frequently encountered only in *C. diphtheriae* and *C. ulcerans*. In *C. diphtheriae*, this gene is carried on a temperate phage that has integrated into the chromosome of a large variety of sublineages (13, 14). In *C. ulcerans*, in addition to a lysogenic phage, the *tox* gene can be carried on a pathogenicity island (15). Strains that carry the *tox* gene generally produce the diphtheria toxin *in vitro*, but a small fraction of them (~10 to 15%) do not, due to disruptions of the *tox* gene; these are called nontoxigenic, *tox*-bearing (NTTB) strains and are observed both for *C. diphtheriae* and for *C. ulcerans* (and for all *C. silvaticum* strains).

Although classically defined as a respiratory infection caused by toxigenic *C. diphtheriae*, diphtheria is sometimes defined more broadly as any infection (respiratory, cutaneous, or other) potentially leading to manifestations due to the production of the diphtheria toxin by species of the CdSC (https://www.ecdc.europa.eu/en/diphtheria). Following recent taxonomic updates, diphtheria may be defined more broadly as any infection caused by any isolate of the CdSC, irrespective of toxigenic status (16).

The typical clinical expressions of diphtheria in human are as follows: (i) classical respiratory diphtheria with pseudomembranous angina that can provoke a deadly obstruction of upper airways, usually associated with fever and enlarged anterior cervical lymph nodes and edema ("bull neck" appearance); (ii) cutaneous manifestations, with "rolled edge" ulcers usually observed on the limbs and that can be covered by a grayish pseudomembrane; and (iii) toxigenic complications such as polyneuropathy or myocarditis (17, 18). Seroprevalence studies in humans demonstrated that up to 82% of the population has titers of anti-diphtheria toxin antibodies below the limit of protection and that the proportion of nonprotected persons increases with age (19, 20).

Before 1999, no cases of *C. ulcerans* were reported in France (21), but between 2002 and 2013, 28 cases of toxigenic *C. ulcerans* were reported and between January 2018 and August 2019, 11 further human clinical cases were described (22); most of the cases were indigenous, and a similar pattern of increased reporting of *C. ulcerans* was found for the United Kingdom and Germany (2, 3, 5, 23).

While the transmission of *C. diphtheriae* is essentially interhuman, *C. ulcerans* is a zoonotic pathogen (2, 24). No transmission of *C. ulcerans* among humans was reported since its description in 1995; however, the possibility of person-to-person transmission cannot be totally excluded (25). Animals from which *C. ulcerans* is isolated can be asymptomatic carriers but can also present clinical symptoms, such as ulcerative dermatitis and chronic rhinitis (26–29). Horses can also carry *C. ulcerans* and sometimes show signs of respiratory diphtheria (30).

Similarly, *C. rouxii* may also be zoonotic: no interhuman transmission has been reported yet, and in addition to human cases, it has been so far identified in dogs, cats, and a fox (10, 31–33). To our knowledge, strains of *C. belfantii* have been isolated only from human respiratory samples.

**TABLE 1** Overview of samples screened and sources of isolates of corynebacteria of the diphtheriae species complex (*CdSC*)

| Clinical source | No. (%) of samples screened | No. of *C. ulcerans* isolates | | No. of *C. rouxii* isolates (all *tox* negative) | No. of *C. diphtheriae* isolates *tox* positive | Total no. (%) of CdSC isolates |
| | | *tox* positive | *tox* negative | | | |
|---|---|---|---|---|---|---|
| Dog | | | | | | |
| Ear swab | 9,439 (52) | 3 | 6 | 8 | 0 | 17 (0.2) |
| Respiratory tract sample | 1,037 (6) | 1 | 3 | 0 | 0 | 4 (0.4) |
| Wound/skin swab | 2,834 (15) | 7 | 8 | 1 | 0 | 16 (0.6) |
| Cat | | | | | | |
| Ear swab | 1,758 (10) | 1 | 2 | 1 | 0 | 4 (0.2) |
| Respiratory tract sample | 1,191 (7) | 6 | 4 | 0 | 0 | 10 (0.8) |
| Wound/skin swab | 1,027 (6) | 2 | 3 | 1 | 0 | 6 (0.6) |
| | | | 1 (cystitis) | | | |
| Horse | | | | | | |
| Ear swab | 5 (0.03) | 0 | 0 | 0 | 0 | 0 (0) |
| Respiratory sample | 111 (0.6) | 0 | 0 | 0 | 0 | 0 (0) |
| Wound/skin swab | 121 (0.7) | 0 | 0 | 0 | 2 (dermatitis/ conjunctivitis) | 2 (1.6) |
| Rat | | | | | | |
| Ear swab | 6 (0.03) | 0 | 0 | 0 | 0 | 0 (0) |
| Respiratory sample | 20 (0.1) | 3 | 0 | 0 | 0 | 3 (15) |
| Wound/skin swab | 10 (0.05) | 0 | 0 | 0 | 0 | 0 (0) |
| Rabbit | | | | | | |
| Ear swab | 116 (0.6) | 0 | 0 | 0 | 0 | 0 (0) |
| Respiratory sample | 289 (1.6) | 1 | 0 | 0 | 0 | 1 (0.3) |
| Wound/skin swab | 127 (0.7) | 0 | 0 | 0 | 0 | 0 (0) |
| Others (goat, sheep, exotics, etc.) | | | | | | |
| Ear swab | 28 (0.2) | 0 | 0 | 0 | 0 | 0 (0) |
| Respiratory sample | 84 (0.5) | 0 | 0 | 0 | 0 | 0 (0) |
| Wound/skin swab | 105 (0.6) | 0 | 0 | 0 | 0 | 0 (0) |
| Total | 18,308 (100) | 24 | 27 | 11 | 2 | 64 (0.3) |

Despite previous reports of animal CdSC infections (for examples, see references 2, 26, 29, and 34), no large case series of such infections, including precise bacterial identification and genotyping, have been reported. Here, we present a case series of CdSC infections in dogs, cats, rabbits, rats, and horses in France over 2 years and report their clinical and microbiological characteristics.

## RESULTS

**Isolation of members of the *C. diphtheriae* species complex in animals in France.** Between August 2019 and August 2021, 18,308 sick animals (dogs, cats, horses, and small mammals) were sampled from veterinary clinics from across metropolitan France. There were 2,732 nasal swabs, 4,224 cutaneous/wound/abscess swabs, and 11,352 auricular swabs (Table 1). A total of 64 consecutive, nonduplicated isolates belonging to the CdSC were identified from these samples (Table 2). Of these, 26 were carriers of the *tox* gene (coding for the diphtheria toxin), including 24 *C. ulcerans* and 2 *C. diphtheriae* isolates. Of the 38 nontoxigenic CdSC isolates, 27 belonged to *C. ulcerans* and the 11 remaining ones were *C. rouxii*. No nontoxigenic *C. diphtheriae*, *C. pseudotuberculosis*, or *C. belfantii* isolates were detected.

The production of the diphtheria toxin was assessed for *tox* gene-bearing isolates. Both toxin gene-carrying *C. diphtheriae* strains were positive in the Elek test. Among the 24 *tox*-positive *C. ulcerans* isolates, one strain (FRC0895) was not available for testing, and 20 of the 23 tested isolates (87.0%) had a positive result by the Elek test, including 3 isolates with a weakly positive Elek test result. These three isolates belonged to a single genetic subtype, sequence type 325 (ST325). Three *tox*-positive *C. ulcerans* isolates (2 of ST331 and 1 of

**TABLE 2** Strain characteristics[a]

| Patient no. | French reference center ID of isolate | Animal species | Age (yrs) | Breed | Sex | French administrative department[d] | Monoinfection | Symptoms | Species | Toxin gene | Elek test | Sequence type | Antibiotic susceptibility test result for indicated group and drug[b] | | | | | | |
|---|---|---|---|---|---|---|---|---|---|---|---|---|---|---|---|---|---|---|---|
| | | | | | | | | | | | | | Aminoglycosides, gentamicin | Beta-lactams, amoxicillin | Macrolides Erythromycin | Spiramycin | Azithromycin | Tetracyclines, tetracycline and doxycycline | Folate pathway inhibitors, trimethoprim-sulfamethoxazole |
| 1 | FRC0804 | Cat | 11 | Domestic shorthair | F | 57 | No | Chronic rhinitis | C. ulcerans | Positive | Positive | ST514 | S | S | S | S | S | S | S |
| 3 | FRC0847 | Cat | 7 | Domestic shorthair | F | 26 | Yes | Chronic rhinitis | C. ulcerans | Positive | Weakly Positive | ST325 | S | S | S | S | S | S | S |
| 5 | FRC0886 | Rat | 0.9 | ND | ND | 40 | No | Chronic rhinitis | C. ulcerans | Positive | Positive | ST332 | I | S | S | S | NA | S | S |
| 6 | FRC0895 | Rat | 1 | ND | M | 40 | No | Chronic rhinitis | C. ulcerans | Positive | NA | NA (contaminated) | S | S | S | NA | NA | S | S |
| 7 | FRC0897 | Cat | ND | Domestic shorthair | M | 41 | Yes | Chronic rhinitis + FIV | C. ulcerans | Positive | Positive | ST757 | S | S | S | S | S | S | S |
| 8 | FRC0916 | Dog | 6 | Mastiff | M | 78 | No | Sore/abscess | C. ulcerans | Positive | Positive | ST325 | S | S | S | S | S | S | S |
| 9 | FRC0922 | Cat | 1 | Domestic shorthair | F | 64 | No | Dermatitis | C. ulcerans | Positive | Positive | ST699 | S | S | S | S | S | S | S |
| 10 | FRC0929 | Dog | 3 | Basset Bleu De Gascogne | F | 71 | Yes | Sore | C. ulcerans | Positive | Positive | ST328 | S | S | S | S | S | S | S |
| 11 | FRC1009 | Dog | 3 | French bulldog | F | 17 | Yes | Chronic otitis | C. ulcerans | Positive | Positive | ST331 | S | S | S | S | S | S | S |
| 12 | FRC1027 | Cat | 8 | Domestic shorthair | M | 60 | No | Chronic otitis | C. ulcerans | Positive | Weakly Positive | ST325 | S | S | S | S | S | S | S |
| 13 | FRC1031 | Dog | 5 | German shepherd | M | 16 | No | Chronic otitis | C. ulcerans | Positive | Positive | ST331 | S | S | S | S | S | S | S |
| 14 | FRC1067 | Dog | 2 | Belgian shepherd | M | 62 | Yes | Abscess | C. ulcerans | Positive | Positive | ST331 | S | S | S | S | S | S | S |
| 15 | FRC1084 | Dog | 9 | X° Lhasa Apso | F | 57 | No | Dermatitis | C. ulcerans | Positive | Positive | ST328 | I | S | S | S | S | S | S |
| 16 | FRC1089 | Dog | 7 | Chow chow | F | 06 | No | Chronic otitis | C. ulcerans | Positive | Weakly Positive | ST325 | I | S | S | S | S | S | S |
| 17 | FRC1102 | Cat | 13 | Domestic shorthair | M | 13 | No | Chronic rhinitis | C. ulcerans | Positive | Positive | ST325 | S | S | S | S | S | S | S |
| 18 | FRC1111 | Cat | 8 | Domestic shorthair | M | 77 | No | Sore | C. ulcerans | Positive | Positive | ST337 | S | S | S | S | S | S | S |
| 19 | FRC1115 | Dog | 8 | Small Bernese hound | M | 07 | No | Dermatitis | C. ulcerans | Positive | Positive | ST690 | I | S | S | S | S | S | S |
| 20 | FRC1123 | Dog | 7 | Jack Russell terrier | M | 42 | No | Sore | C. ulcerans | Positive | Negative | ST331 | I | S | S | S | S | S | S |
| 21 | FRC1127 | Rabbit | 0.8 | ND | F | 60 | No | Chronic rhinitis | C. ulcerans | Positive | Negative | ST358 | S | S | S | S | S | S | S |
| 22 | FRC1128 | Dog | 1 | Cane Corso | F | 12 | Yes | Chronic rhinitis | C. ulcerans | Positive | Positive | ST325 | S | S | S | S | S | S | S |
| 23 | FRC1137 | Rat | 1 | ND | M | 25 | Yes | Chronic rhinitis | C. ulcerans | Positive | Positive | ST332 | S | S | S | S | S | S | S |
| 24 | FRC1148 | Cat | 15 | Domestic shorthair | F | 78 | No | Chronic rhinitis | C. ulcerans | Positive | Negative | ST331 | I | S | S | S | S | S | S |
| 25 | FRC0855 | Cat | 5 | Domestic shorthair | F | 95 | Yes | Chronic rhinitis | C. ulcerans | Negative | NR | ST325 | S | S | S | S | S | S | S |
| 26 | FRC0856 | Dog | ND | German shepherd | ND | ND | No | Chronic otitis | C. ulcerans | Negative | NR | ST720 | I | S | S | S | S | S | S |
| 28 | FRC0915 | Cat | 1 | Maine Coon | F | 46 | No | Chronic rhinitis | C. ulcerans | Negative | NR | ST339 | S | S | S | S | S | S | S |
| 29 | FRC0921 | Dog | 5 | French mastiff | M | 57 | No | Dermatitis | C. ulcerans | Negative | NR | ST699 | S | S | S | S | S | S | S |
| 30 | FRC0971 | Dog | ND | Griffon | M | 45 | No | Chronic otitis | C. ulcerans | Negative | NR | ST339 | S | S | S | S | S | S | S |
| 31 | FRC0973 | Dog | 9 | Beauceron | F | 47 | No | Chronic otitis | C. ulcerans | Negative | NR | ST760 | S | S | S | S | S | S | S |
| 32 | FRC0981 | Dog | 4 | German shepherd | M | 77 | No | Chronic rhinitis | C. ulcerans | Negative | NR | ST339 | S | S | S | S | S | S | S |
| 34 | FRC0990 | Dog | 6 | German shepherd | M | 69 | No | Chronic rhinitis | C. ulcerans | Negative | NR | ST339 | S | S | S | S | S | S | S |
| 36 | FRC1028 | Dog | 0.3 | Great Dane | F | 77 | No | Dermatitis | C. ulcerans | Negative | NR | ST690 | S | S | S | S | S | S | S |
| 37 | FRC1032 | Cat | 11 | ND | F | 35 | Yes | Cystitis | C. ulcerans | Negative | NR | ST690 | S | S | S | S | S | S | S |
| 38 | FRC1047 | Cat | 17 | Domestic shorthair | F | 64 | No | Sore | C. ulcerans | Negative | NR | ST428 | S | S | S | S | S | S | S |
| 39 | FRC1054 | Cat | 7 | Domestic shorthair | M | 92 | No | Sore | C. ulcerans | Negative | NR | ST339 | S | S | R | R | R | S | S |
| 41 | FRC1110 | Dog | 6 | Cocker spaniel | M | 71 | No | Chronic otitis | C. ulcerans | Negative | NR | ST339 | S | S | S | S | S | S | S |
| 42 | FRC1114 | Cat | 13 | Birman | F | 16 | Yes | Chronic rhinitis | C. ulcerans | Negative | NR | ST325 | I | S | S | S | S | S | S |

**TABLE 2** (Continued)

| Patient no. | French reference center ID of isolate | Animal species | Age (yrs) | Breed | Sex | French administrative department[d] | Monoinfection | Symptoms | Species | Toxin gene | Elek test | Sequence type | Antibiotic susceptibility test result for indicated group and drug[b] | | | | | | |
|---|---|---|---|---|---|---|---|---|---|---|---|---|---|---|---|---|---|---|---|
| | | | | | | | | | | | | | Aminoglycosides, gentamicin | Beta-lactams, amoxicillin | Macrolides, Erythromycin | Spiramycin | Azithromycin | Tetracyclines, tetracycline and doxycycline | Folate pathway inhibitors, trimethoprim-sulfamethoxazole |
| 43 | FRC1134 | Cat | 6 | Porcelaine | F | 45 | No | Chronic rhinitis | C. ulcerans | Negative | NR | ST339 | S | S | S | S | S | S | S |
| 44 | FRC1146 | Dog | 5 | Porcelaine | F | 38 | No | Sore | C. ulcerans | Negative | NR | ST339 | I | S | S | S | S | S | S |
| 45 | FRC1153 | Dog | 4 | Great Dane | F | 59 | No | Sore | C. ulcerans | Negative | NR | ST339 | R | S | S | S | S | S | S |
| 46 | FRC1164 | Dog | 6 | Great Dane | M | 57 | No | Dermatitis | C. ulcerans | Negative | NR | ST327 | S | S | S | S | S | S | S |
| 47 | FRC1165 | Cat | 7 | Domestic shorthair | F | 18 | No | Sore | C. ulcerans | Negative | NR | ST325 | S | S | S | S | S | S | S |
| 48 | FRC1170 | Dog | 9 | German shepherd | M | 13 | No | Chronic rhinitis | C. ulcerans | Negative | NR | ST339 | S | S | S | S | S | S | S |
| 49 | FRC1194 | Dog | 9 | German shepherd | M | 70 | No | Chronic otitis | C. ulcerans | Negative | NR | ST339 | S | S | S | S | S | S | S |
| 50 | FRC1193 | Cat | 7 | Rex Devon | M | 67 | No | Chronic otitis | C. ulcerans | Negative | NR | ST325 | S | S | S | — | S | S | S |
| 51 | FRC0848 | Cat | 13 | Domestic shorthair | M | 17 | No | Chronic otitis | C. ulcerans | Negative | NR | ST720 | S | S | S | S | S | S | S |
| 52 | FRC1017 | Horse | 30 | ND | F | 33 | No | Conjunctivitis | C. diphtheriae | Positive | Positive | ST669 | S | S | S | S | S | S | S |
| 53 | FRC1117 | Horse | 8 | ND | M | 40 | No | Dermatitis | C. diphtheriae | Positive | Positive | ST59 | S | S | S | S | S | S | S |
| 54 | FRC0802 | Dog | 9 | Drahthaar | M | 37 | No | Sore | C. rouxii | Negative | NR | ST694 | I | S | S | S | S | S | S |
| 55 | FRC0810 | Dog | 9 | ND | F | 84 | No | Chronic otitis | C. rouxii | Negative | NR | NA (*rpoB* truncated) | I | S | S | S | S | S | S |
| 56 | FRC0845 | Dog | ND | Labrador | ND | ND | No | Chronic otitis | C. rouxii | Negative | NR | ST714 | S | S | S | S | S | S | S |
| 57 | FRC0835 | Cat | ND | Domestic shorthair | ND | ND | Yes (+ *Malassezia*) | Chronic otitis | C. rouxii | Negative | NR | ST713 | S | S | S | S | S | S | S |
| 58 | FRC0926 | Cat | 1 | Domestic shorthair | M | 47 | No | Sore | C. rouxii | Negative | NR | ST537 | I | S | S | S | S | S | S |
| 59 | FRC0976 | Dog | 12 | German shepherd | F | 47 | No | Chronic otitis | C. rouxii | Negative | NR | ST762 | S | S | S | S | S | S | S |
| 60 | FRC1088 | Dog | 6 | American Staffordshire bull terrier | M | 84 | No | Chronic otitis | C. rouxii | Negative | NR | ST537 | I | S | S | S | S | S | S |
| 61 | FRC1100 | Dog | 13 | X Beagle | F | 17 | No | Chronic otitis | C. rouxii | Negative | NR | ST780 | S | S | S | S | S | S | S |
| 62 | FRC1150 | Dog | 7 | X Griffon | F | 64 | No | Chronic otitis | C. rouxii | Negative | NR | ST537 | S | S | S | S | S | S | S |
| 63 | FRC1159 | Dog | 9 | Labrador | M | 33 | No | Chronic otitis | C. rouxii | Negative | NR | ST537 | S | S | S | S | S | S | S |
| 64 | FRC1172 | Dog | 3 | Griffon | M | 65 | No | Chronic otitis | C. rouxii | Negative | NR | ST537 | S | S | S | S | S | S | S |

[a]ID, identifer; F, female; M, male; FIV, feline immunodeficiency virus; ND, not documented; NR, not documented; NA, not available; S, susceptible; I, intermediate; R, resistant.

[b]breakpoints: gentamicin, S ≥ 18 and R < 16; amoxicillin, S ≥ 23 and R < 16; erythromycin, S ≥ 22 and R < 19; azithromycin, S ≥ 24 and R < 19; spiramycin, S ≥ 22 and R < 17; tetracycline and doxycycline, S ≥ 19 and R < 17; trimethoprim-sulfamethoxazole, S ≥ 16 and R < 10.

[c]X, mixed breed.

[d]https://en.wikipedia.org/wiki/Departments_of_France.

ST358) were negative by the Elek test, hence corresponding to nontoxigenic, *tox* gene-bearing (NTTB) strains.

Of the 64 infections with CdSC isolates, 37 were in dogs, 21 in cats, 3 in rats, and 2 in horses and 1 was in a rabbit (Tables 1 and 2). The ages of dogs and cats ranged from 1 year to 13 years. There was no statistically significant difference in prevalence of CdSC isolates among the different age groups. The most frequent breed among the *C. ulcerans* dog cases was the German shepherd (9 of 28 dogs of this breed). Dogs belonging to this breed were strongly associated with infections caused by *C. ulcerans*: there were 556 German shepherds out of 13,310 dogs ($P < 0.00001$).

**Antimicrobial susceptibility of CdSC isolates.** *C. ulcerans* isolates were susceptible to most antibiotics (Table 2). Spiramycin was tested in 50 isolates, following the publication of Abbott et al. in 2020 (29); all were susceptible, except 1 resistant and 1 susceptible at a higher concentration (previously "intermediate"). Azithromycin was tested in 49 samples, and only 1 isolate (*C. ulcerans* FRC1054) was resistant; this isolate was also resistant to erythromycin and spiramycin. Nine *C. ulcerans* isolates tested intermediate for gentamicin and 1 was resistant, whereas the 42 other ones were susceptible. The *C. ulcerans* isolates were also susceptible to erythromycin in all but one case and were all susceptible to tetracyclines and trimethoprim-sulfamethoxazole.

The two *C. diphtheriae* isolates were susceptible to all antibiotics tested, including clindamycin. The *C. rouxii* isolates were susceptible to amoxicillin, tetracyclines, trimethoprim-sulfamethoxazole, erythromycin, spiramycin, azithromycin, and clindamycin, but only 7 isolates were susceptible to gentamicin, the 4 others being intermediate.

**Genotyping of isolates using MLST.** Multilocus sequence typing (MLST) analysis showed a wide diversity of *C. ulcerans* isolates, with 15 distinct STs (Fig. 1 and Table 2). *C. rouxii* was also genetically heterogeneous, with 7 STs, and the two *C. diphtheriae* isolates belonged to ST59 and ST699 (Fig. 1). Six *C. ulcerans* STs and one *C. rouxii* ST comprised more than one isolate. For these STs, geographical provenance and animal source were heterogeneous, indicating spread across localities and host species (see Fig. S1 in the supplemental material). Two *C. ulcerans* STs (ST325 and ST690) comprised both *tox*-positive and *tox*-negative isolates (Fig. 1), implying that the loss or gain of the *tox* gene occurred within these lineages. We noted that *C. rouxii* isolates originated mainly from southwestern France (Fig. 2 and Fig. S1).

**Clinical characteristics of toxigenic *C. ulcerans* isolates.** Eleven of the 24 toxigenic *C. ulcerans* isolates were isolated from nasal swabs. They were taken from 6 cats with chronic rhinitis and 3 rats, 1 dog, and 1 rabbit. Two of the rats were bought from the same pet shop, and they were both coinfected with *Staphylococcus aureus*. Euthanasia was chosen because of the zoonotic potential. Four other nasal swabs also harbored microorganisms other than *C. ulcerans*, including *Pasteurella multocida*, *Escherichia coli*, *Bacteroides* spp., *Stenotrophomonas maltophilia*, and *Staphylococcus pseudointermedius*.

The second most frequent isolation source was skin (9 of 24 cases) and included wounds, pyoderma, and abscesses. Seven of these *C. ulcerans* were sampled from dogs and two from cats. In 4 of the 9 skin cases, *C. ulcerans* was the only microorganism recovered from the samples, while in 5 cases there was a coinfection; organisms encountered were *Pasteurella multocida*, *Pasteurella canis*, *Proteus mirabilis*, *Staphylococcus aureus*, *Staphylococcus pseudointermedius*, *Streptococcus canis*, and *Pseudomonas aeruginosa*.

Last, there were four ear infections by toxigenic *C. ulcerans*. These were observed in three dogs and one cat. In 3 cases, otitis was also associated with at least another microorganism (*Proteus mirabilis*, *Pseudomonas aeruginosa*, *Streptococcus canis*, or *Streptococcus dysgalactiae*).

Three of the 24 animals infected by a toxigenic *C. ulcerans* strain had a bacteriological follow-up examination. Two cats, patients 3 and 4 (Table 2), had rhinitis with *C. ulcerans* in pure culture and tested positive again at control screenings 1 month and 18 days after the initial visit, respectively. Patient 3 had been treated for 10 days with amoxicillin-clavulanate, without success. No further control was performed. In patient 4, a 4-week cure of cefovecin did not eliminate the toxigenic *C. ulcerans*. However, a 1-month treatment with

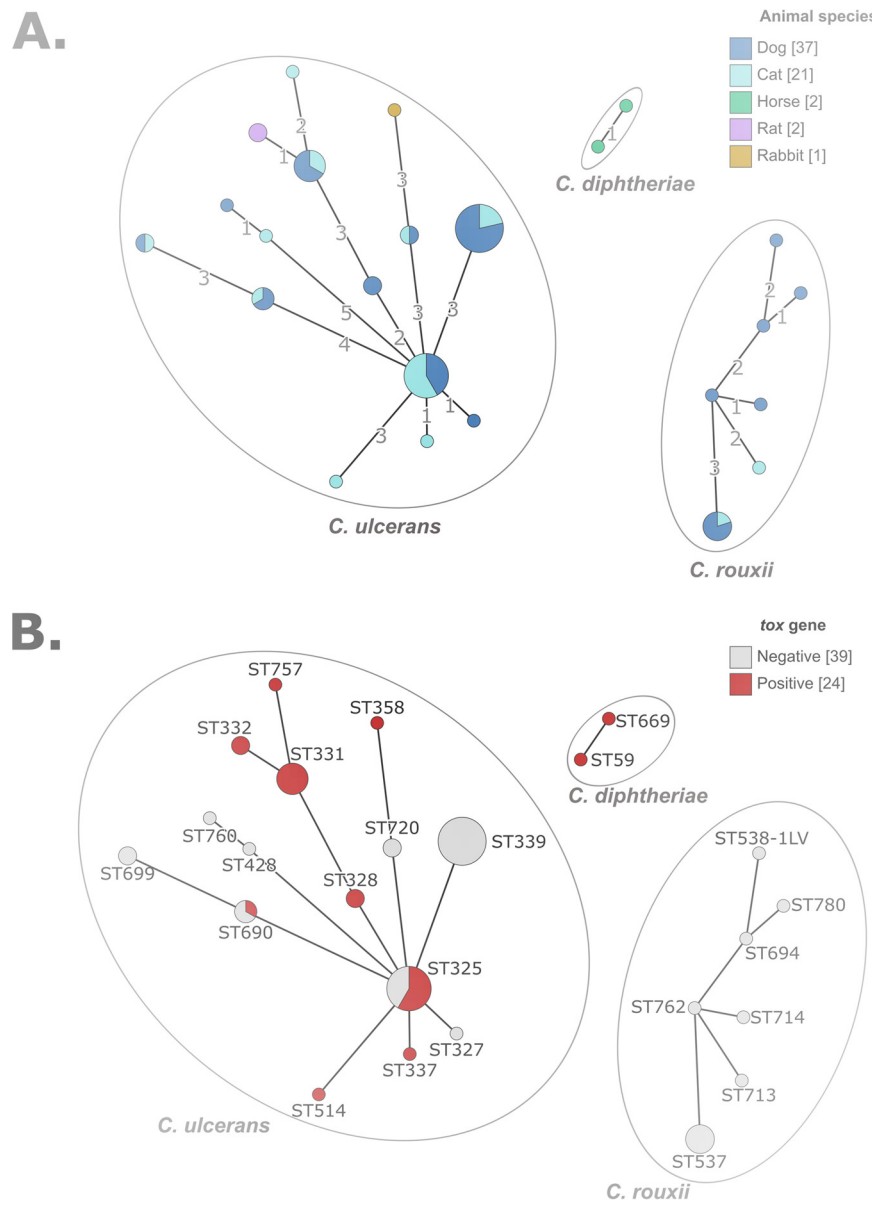

**FIG 1** MLST diversity of CdSC isolates from pets. (A) Minimum spanning tree of 7-gene MLST profiles, colored in function of animal host species. (B) Same as panel A, colored by diphtheria toxin gene presence. The graphs were obtained using the GrapeTree tool, which is plugged onto the BIGSdb platform (https://bigsdb.pasteur.fr/cgi-bin/bigsdb/bigsdb.pl?db=pubmlst_diphtheria_isolates&page=plugin&name=GrapeTree).

amoxicillin-clavulanate, to which the isolate was susceptible, was followed by a negative result for *C. ulcerans* on the control sample 4 months later.

One rabbit (patient 21) with *C. ulcerans* and *Pasteurella multocida* coinfection was still positive for *C. ulcerans* and *P. multocida* despite treatment with marbofloxacin for 7 days and then trimethoprim-sulfamethoxazole for 10 more days. No further control was performed.

**Clinical characteristics of cases with nontoxigenic *C. ulcerans* isolates.** The 27 nontoxigenic *C. ulcerans* isolates were retrieved from 6 cases of dermatitis (6 dogs), 7 cases of rhinitis (4 cats and 3 dogs), 8 cases of otitis (6 dogs and 2 cats), 1 case of cystitis (cat), and 5 cases of nonhealing purulent wounds (3 cats and 2 dogs). In total, there were 10 cats and 17 dogs. Three monoinfections were observed and concerned two cats with chronic rhinitis and one cat with cystitis.

The other infections were polymicrobial. Regarding the otitis cases, the polyinfections consisted of coinfection with *Streptococcus canis* (in 3 cases), *Pseudomonas aeruginosa*

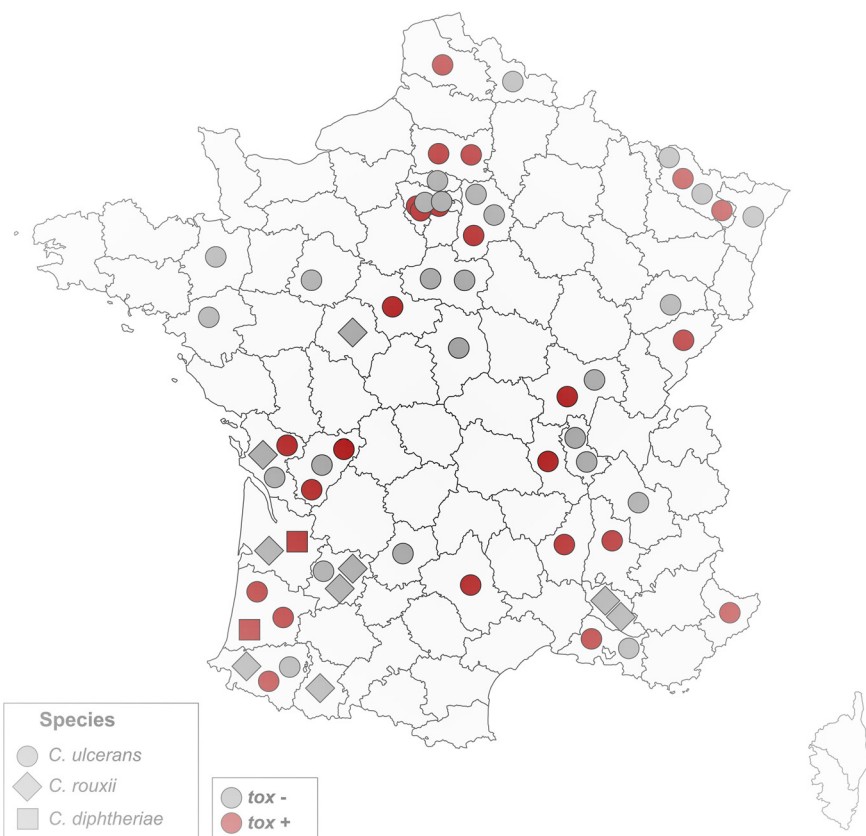

**FIG 2** Geographic distribution of CdSC infection cases in pets. The 51 *Corynebacterium ulcerans*, 2 *C. diphtheriae*, and 9 *C. rouxii* are displayed within their administrative department of origin, with a species-specific symbol. *tox* gene-bearing strains are indicated in red and non-*tox* gene-bearing strains in gray. For two cases, location was not available, and they are therefore not represented here.

(*n* = 2), *Proteus mirabilis* (*n* = 2), *Staphylococcus pseudointermedius* (*n* = 2), and *Klebsiella pneumoniae* (*n* = 1). The polyinfections in the cases of rhinitis were coinfections with *Staphylococcus intermedius* (*n* = 2), *Pasteurella multocida* and *Bordetella bronchiseptica* (*n* = 1), *Klebsiella oxytoca* (*n* = 1), and *Staphylococcus aureus* (*n* = 1). The dermatitis polyinfections were coinfections with *S. pseudointermedius* (*n* = 5), *P. mirabilis* (*n* = 3), and *Pasteurella canis* (*n* = 1). Last, polyinfections of sores comprised *Escherichia coli* (*n* = 2), *S. aureus/pseudointermedius* (1 case each), *Pasteurella dagmatis/multocida* (1 case each), and *Streptococcus dysgalactiae* (*n* = 1).

There was only one follow-up available (patient 25). It was for a 5-year-old cat with chronic rhinitis and a *C. ulcerans* monoinfection. This patient remained positive for nontoxigenic *C. ulcerans* for 6 months. In between, the cat was treated with doxycycline to which the strain tested susceptible in the 2 antibiotic susceptibility tests before and after treatment.

**Clinical characteristics of cases with *C. diphtheriae* and *C. rouxii* isolates.** The two cases of *C. diphtheriae* were toxigenic and consisted of conjunctivitis and pastern dermatitis in two horses. Both cases were polymicrobial infections, in which *S. dysgalactiae* (conjunctivitis) and *S. aureus* and *Enterobacter cloacae* (pastern dermatitis) were also found.

*C. rouxii* infections were observed in 9 dogs and 2 cats and were all nontoxigenic. They consisted of 9 otitis cases (8 dogs and 1 cat) and two nonhealing wounds (one dog and one cat). *C. rouxii* was identified retrospectively by MLST, as this species was only recognized as a separate species during the study period (in 2020), and as matrix-assisted laser desorption ionization–time of flight mass spectrometry (MALDI-TOF MS)

still identifies *C. rouxii* as *C. diphtheriae*, based on the MALDI Biotyper Compass database version 4.1.

Almost all *C. rouxii* infections were polymicrobial infections. There was only one monobacterial infection (ear infection of a cat), though *Malassezia* was also identified (Table 2). The most frequent (*n* = 5) coinfecting bacterium was *P. aeruginosa*, as expected in cases of otitis. Other coinfecting bacteria were *P. mirabilis* (*n* = 3), *Staphylococcus* sp. (*n* = 2), *S. canis* (*n* = 2), *Corynebacterium amycolatum* (*n* = 1), *Citrobacter koseri* (*n* = 1), and anaerobic bacteria (*n* = 1). The age of the infected animals ranged from 1 year to 13 years, with no statistically significant difference between age groups.

## DISCUSSION

We report on the occurrence and clinical and microbiological characteristics of infections caused in companion animals by corynebacteria of the *diphtheriae* species complex (CdSC) in France. Although *C. ulcerans* was the most frequent species, *C. rouxii* and, more rarely, *C. diphtheriae* were also found.

All isolates initially identified as *C. diphtheriae* from dogs and cats were in fact *C. rouxii*, as identified by MLST gene sequences. This important observation suggests that this novel species has been overlooked, and it in fact appears more prevalent than *C. diphtheriae* itself in animal infections. As all *C. rouxii* infections, mostly of ears, were coinfections with other bacteria or the yeast *Malassezia*, this species may represent a commensal in dogs and cats. So far, all *C. rouxii* isolates are *tox* negative, with the exception of isolates from two cats from the United States (31), but in these two isolates, the *tox* gene was disrupted and therefore the toxin was not produced. Previously, *C. rouxii* has been isolated from cutaneous infections, vasculitis, and peritonitis in humans and a purulent orbital cellulitis in a dog (10). Our study corresponds to the largest group of infections with *C. rouxii* reported so far. We suggest that *C. rouxii* may represent a novel zoonotic pathogen, as pets may clearly serve as a reservoir for these human infections. However, so far, no case of transmission of *C. rouxii* between humans and animals has been documented. *C. rouxii* was more frequently isolated in southwestern France (Fig. 2) and was genetically diverse (Fig. 1 and Fig. S1). Although the sampling is still limited, this may reflect the existence of local conditions that favor the infections of dogs or cats by *C. rouxii* in this part of France.

Toxigenic *C. diphtheriae* was isolated from 2 horses. Mixed wound infections in horses have been previously described (34, 35), and colonization with *C. diphtheriae* was reported in 6.9% of slaughter horses in Romania (36). The pathogenic potential of *C. diphtheriae* in horses is questionable given its report from polymicrobial infections or asymptomatic carriage. Given the well-established human-to-human transmission of *C. diphtheriae*, and possible asymptomatic carriage, the possibility should be considered that the detection of this pathogen in horses corresponds to reverse zoonosis.

*C. ulcerans* is now well established as a zoonotic member of the CdSC. Whereas no human-to-human transmission was reported, human cases have often been associated with animal contacts, and in several cases the genetic fingerprinting of isolates supported an epidemiological link between the animal and human isolates, strongly establishing the zoonotic character of *C. ulcerans* (24, 26, 37). The emergence of *C. ulcerans* has been noted in France, Germany, and the United Kingdom (3–5, 38, 39).

This is the first study based on the systematic analysis of a very large animal cohort (18,308 samples), which provides data on the frequency of *C. ulcerans* in various types of clinical samples from animals. Abbott and colleagues (29) found 7 *C. ulcerans* isolates among 804 nasal samples (0.87%; 3 samples from 668 dogs and 4 samples from 64 cats), whereas Katsukawa et al. (28) found 44 *C. ulcerans* isolates in 583 pharyngeal samples of dogs (7.5%). In this study, we found 51 *C. ulcerans* isolates, including 24 *tox*-positive ones, in 18,308 clinical samples (0.27%), mostly in nasal swabs (11 *tox*-positive and 7 tox-negative samples).

Given the wide geographic distribution of cases (Fig. 2 and Fig. S1), the reporting of *C. ulcerans* in this study does not seem affected by a bias caused by local events of transmission. In addition, the genetic diversity of *C. ulcerans* isolates indicates that the reporting of this pathogen cannot be attributed to the clonal spread of a single emerging

strain of *C. ulcerans*. Whether the increase in cases observed over the two last decades is due to changes in diagnostic practice (i.e., MALDI-TOF mass spectrometry) or increasing awareness of the necessity to report and test for the presence of the diphtheria toxin, rather than a real epidemiological phenomenon of emergence, remains unclear. Before 2019, *C. ulcerans* isolates of animal origin were sent only sporadically to the national reference laboratory. Awareness of this zoonotic bacterium remains low among veterinarians and veterinary laboratories, among which it is often considered a commensal of animals.

*C. ulcerans* is widely distributed, having been reported from multiple animal species (2). Whether pet animals represent a natural reservoir or (more probably) are themselves contaminated by *C. ulcerans* from other animal species or sources is an important question. Toxigenic *C. ulcerans* infections were reported for wildlife carnivores such as foxes, otters, and owls (40) and insectivores (hedgehogs and Japanese shrew owls) (41), whereas nontoxigenic *C. ulcerans* infections seem to be more frequent in omnivores (wild boars) and herbivores (roe deer) (38). Only wild animals with major symptoms are diagnosed, while the asymptomatic carriership is rarely investigated (42). Transmission by predatory hunting was supported by a study showing high serum diphtheria antitoxin titers in hunting dogs (43). A reservoir of symptomatic or asymptomatic carriers in small wild mammals such as herbivores, lagomorphs, or rodents, which may be prey to dogs and cats, should be further investigated. Here, we report 4 cases of toxigenic *C. ulcerans* in rats or rabbits. Although the cases described are from symptomatic animals, an asymptomatic carrier state of *C. ulcerans* in dogs, cats, horses, rodents, and wildlife animals has been described in other studies (24, 37, 41, 43, 44). Future studies should investigate the presence of CdSC in healthy animals to better understand and control its transmission to pets.

There is evidence for a pathogenic potential of *C. ulcerans* in animals (2), and experimental intranasal and intravenous infection studies in mice showed the pathogenic capacity of *C. ulcerans*, irrespective of toxigenicity (45, 46). In this study, *C. ulcerans* was often isolated from nasal swabs of cats with chronic rhinitis. While acute feline upper respiratory infections are mostly caused by viruses (feline herpesvirus 1 and feline calicivirus) (47) and are most of the time self-limiting, chronic rhinitis is more concerning. In general, bacteriological examinations performed on nasal swabs are often not useful because of the presence of commensal flora and only a few bacteria are considered primary pathogens (47). Toxigenic *C. ulcerans* may be implicated in the clinical manifestation of rhinitis. Importantly, in 7 cases of rhinitis, *C. ulcerans* was the only infective agent retrieved from the nasal swabs, a fact that strongly suggests a primary pathogenic nature of *C. ulcerans* in cats, consistent with a previous study (29). A limitation of the present study is that it was performed retrospectively; hence, the final diagnosis as well as the complete clinical picture was not available.

The pathogenicity of nontoxigenic strains is not well understood. Here, a few monoinfections by nontoxigenic *C. ulcerans* (2 cases of rhinitis and 1 cystitis) are reported. The infections could be favored by other virulence factors of *C. ulcerans*, such as phospholipase D (2).

Large dog breeds were disproportionally affected by *C. ulcerans*, and 32% of them were German shepherds. This interesting observation might be explained by a closer contact with small mammals, as large dogs are generally kept outdoors. But it could also reflect a breed predisposition, as German shepherd dogs are more susceptible to pyoderma (48) and other bacterial infections, for example, by *Ehrlichia canis* (49). Six German shepherds in this study presented with pyoderma or otitis. Further studies will be needed to investigate a possible dog breed predisposition to *C. ulcerans*.

In at least 3 cases of toxigenic *C. ulcerans* and 1 case of nontoxigenic *C. ulcerans* (patient numbers 3, 4, 21, and 25), the infection persisted, despite antibiotic susceptibility, as reported in previous works (26, 29, 37). One possibility is that chronic infections might involve some intracellular bacteria (50), which would be more accessible to macrolides, which can reach higher intracellular concentrations than beta-lactams. Systemic infections by *C. diphtheriae*

suggest that this pathogen is not only able to attach to host epithelial cells but also able to gain access to deeper tissues and to live intracellularly (51). Guidelines for *Corynebacterium* in humans recommend the use of beta-lactams or macrolides. Yet, while chronic human carriers can be treated with azithromycin or rifampicin, the use of the latter is forbidden in the veterinary field.

Recent data suggest the use of spiramycin in cases of animal infection by *C. ulcerans*. A 10-day course of the combination of spiramycin and metronidazole was successful in clearing *C. ulcerans* from a dog, and a 6-day course of the same antibiotic combination was successful in cats (18, 29). As spiramycin only exists in a combination with metronidazole, and as metronidazole has side effects (neurotoxicity and effect on the microbiota), azithromycin seems to be a better choice for treatment.

The efficiency of the antibiotic treatment should be verified ideally by 2 samples, taken 5 to 7 days following completion of the antibiotic course. The macrolide agent (azithromycin rather than a spiramycin-metronidazole combination) and the optimal treatment duration should be investigated by future studies.

This work is relevant to the development of veterinarian guidelines in the case of CdSC infections in animals. Currently, in France no recommendations are established around the detection of toxigenic CdSC in animals, either for animal or their human contacts. From a public health perspective, the zoonotic potential of toxigenic *C. diphtheriae* or *C. ulcerans* in chronic skin infections, nonhealing wounds, rhinitis, and otitis justifies investigating these body locations for such pathogens, and these samples should not be considered to reveal only commensal organisms. We suggest that in the case of CdSC detection in animals, the veterinary laboratories should be encouraged to send the samples to a reference laboratory for analysis of the presence of the *tox* gene. This would be more in line with the fact that in the case of diphtheria due to toxigenic *C. ulcerans* in a human, recommendations exist in France that contact animals should be sampled and treated.

Toxigenic corynebacteria in animals are not mandatory to notify, and the costs of treatment, and clearance and sampling of contact animals, clearly represent a limitation for the implementation of follow-up investigations and control measures. Asymptomatic human carriers of toxigenic strains are treated with the same antibiotic regimen as symptomatic cases, with clearance swabs to ensure eradication. From an epidemiological viewpoint, it would be coherent to follow the same practice for animals, and this should be discussed between veterinary and human health institutions.

## MATERIALS AND METHODS

**Inclusions and microbiological characterization.** A total of 18,308 samples originating from sick animals (dogs, cats, horses, and small mammals) from metropolitan France were analyzed during the period from August 2019 to August 2021 (24 months) (Table 1). All samples from which isolates were identified as members of the CdSC were included in the present study. Although *C. pseudotuberculosis* was sometimes recorded, this species was excluded because the shipping and reception process of these samples is known to be biased (samples of livestock are generally sent to state laboratories and only occasionally to private laboratories).

The geographical location of the cases was recorded using the corresponding French administrative department (in metropolitan France, there are 96 geographic and administrative divisions called "départements"). Clinical data (age, sex, and clinical symptoms) were collected using information communicated on the order form.

Sterile swabs with liquid Amies culture medium were used to sample the affected animals and were sent to the laboratory under cooled conditions. There, swabs were plated on solid culture media (Columbia agar supplemented with 5% sheep blood and colistin plus nalidixic acid [CNA]) under an extractor hood. Solid culture media were incubated at 35°C under a 5% enriched $CO_2$ atmosphere for 24 to 48 h. Isolates belonging to the *Corynebacterium diphtheriae* species complex form dry colonies of a dark yellow color on CNA culture medium. The identity of the suspected colonies was confirmed by matrix-assisted laser desorption ionization–time of flight mass spectrometry (MALDI-TOF MS; Bruker Daltonics, Germany). Note that this method identifies *C. diphtheriae* and *C. ulcerans* but does not discriminate well other members of the CdSC from these two species.

Antibiotic susceptibility testing was performed by disk diffusion on Mueller-Hinton culture medium supplemented with 5% blood following the instructions of the French AFNOR standard NF U47-107 (https://www.boutique.afnor.org/fr-fr/norme/nf-u47107/methodes-danalyse-en-sante-animale-guide-de-realisation-des-antibiogrammes-/fa170310/40286). Diameters of the inhibition zone were read and interpreted using software tool SIRWeb (i2A, France). As there are no veterinary breakpoints, the 2013 guidelines from the

human CASFM (https://www.sfm-microbiologie.org/wp-content/uploads/2020/07/CASFM_2013.pdf) were applied. The antibiotic susceptibility testing included antibiotics commonly used in veterinary clinics, including beta-lactams (amoxicillin), tetracyclines (doxycycline and tetracycline), aminoglycosides (gentamicin), macrolides (erythromycin, azithromycin, and spiramycin), and others (trimethoprim-sulfamethoxazole). Recent data suggest that *C. ulcerans* is less susceptible to clindamycin (52). Therefore, clindamycin was tested only for *C. diphtheriae* and *C. rouxii*. As there are no breakpoints for veterinary fluoroquinolones and cefovecin, these antibiotics were not tested.

All CdSC isolates were sent to the National Reference Center for Corynebacteria of the *diphtheriae* complex for confirmation of the identification and for the detection of the diphtheria toxin (*tox*) gene by real-time PCR (53). This multiplex assay consists of amplifying a fragment of the *rpoB* gene with primer sets specific for either (i) the species *C. diphtheriae*, *C. belfantii*, or *C. rouxii* or (ii) the species *C. ulcerans* and *C. pseudotuberculosis*; in addition, a fragment of the *tox* gene is detected. The production of the diphtheria toxin was assessed using the modified Elek test (54).

**MLST.** Isolates were retrieved from −80°C storage and plated on tryptose-casein soy agar for 24 to 48 h. A small amount of bacterial colony biomass was resuspended in a lysis solution (20 mM Tris-HCl [pH 8], 2 mM EDTA, 1.2% Triton X-100, and lysozyme [20 mg/mL]) and incubated at 37°C for 1 h, and DNA was extracted with the DNeasy blood and tissue kit (Qiagen, Courtaboeuf, France) according to the manufacturer's instructions. Multilocus sequence typing (MLST) was performed as previously described (13, 55); alleles and profiles were defined using the BIGSdb-Pasteur platform (https://bigsdb.pasteur.fr/diphtheria).

**Statistical analyses.** Dogs and cats were divided into 3 age groups: under 2 years, 2 to 8 years, and older than 8 years. A few dogs could not be attributed to a group because the information was not available. Pearson's chi-squared statistic was used to compare the different characteristics for the categorical variables. Statistical tests were performed in SPSS Statistics software version 25 (IBM, New York, NY).

**Ethics statement.** Animals were sampled for diagnostic purposes. No ethics approval was requested for this retrospective study.

**Data availability.** MLST sequences and profiles can be accessed at https://bigsdb.pasteur.fr/diphtheria.

## SUPPLEMENTAL MATERIAL

Supplemental material is available online only.

**SUPPLEMENTAL FILE 1**, PDF file, 0.4 MB.

## ACKNOWLEDGMENTS

We thank Stéphanie Gilles from the bacteriology department of Cerba Vet for her support and Annick Carmi-Leroy, Annie Landier and Nathalie Armatys (Institut Pasteur) for technical help.

This work was supported financially by the French Government's Investissement d'Avenir program Laboratoire d'Excellence "Integrative Biology of Emerging Infectious Diseases" (ANR-10-LABX-62-IBEID). This work used computational and storage services provided by the IT department at Institut Pasteur. The National Reference Center for Corynebacteria of the *diphtheriae* complex is supported by Institut Pasteur and Santé publique France (Public Health France). Initial microbiological analyses and logistics for referring isolates to the National Reference Laboratory were supported financially by Cerba Vet. M.H. was supported financially by the Ph.D. grant "Codes4strains" from the European Joint Program One Health, which has received funding from the European Union's Horizon 2020 Research and Innovation Program under grant agreement no. 773830.

We declare that 2 authors (K.M. and G.R.) are employees of Cerba Vet, which performs diagnostic testing on a commercial basis.

K.M. and S.B. conceived, designed, and coordinated the study. G.R. and E.B. performed the microbiological cultures of the isolates and their biochemical and molecular characterizations. M.H. analyzed the sequence data and created the figures. G.A. and K.M. reviewed the clinical source data of the isolates. K.M. and S.B. wrote the initial version of the manuscript. All authors provided input to, read, and approved the final manuscript.

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
