## [Reviewer comments · Microbiology Spectrum]

Microbiology Spectrum

Corynebacterium of the diphtheriae complex in companion animals: clinical and microbiological characterization of 64 cases from France

Kristina Museux, Gabriele Arcari, Guido Rodrigo, Melanie Hennart, Edgar Badell, Julie Toubiana, and Sylvain Brisse

Corresponding Author(s): Sylvain Brisse, Institut Pasteur

Review Timeline:

Submission Date:	January 10, 2023
Editorial Decision:	February 4, 2023
Revision Received:	February 16, 2023
Accepted:	March 7, 2023

Editor: Florence Doucet-Populaire

Reviewer(s): The reviewers have opted to remain anonymous.

Transaction Report:

DOI: <https://doi.org/10.1128/spectrum.00006-23>

February 4, 2023

Dr. Sylvain Brisse
Institut Pasteur
Biodiversity and Epidemiology of Bacterial Pathogens
25-28 rue du Dr Roux
Paris 75724
France

Re: Spectrum00006-23 (Corynebacterium of the diphtheriae complex in companion animals: clinical and microbiological characterization of 64 cases from France)

Dear Dr. Sylvain Brisse:

Link Not Available

Sincerely,

Florence Doucet-Populaire

Journals Department
Reviewer comments:

Reviewer #1 (Comments for the Author):

In this report, Museux et. al, present the results of case studies performed in France over a two-year period that identified possible diphtheria infections in animals. The infections were associated with the presence of bacteria that are members of the Corynebacterium diphtheriae complex (Cdc), which includes the pathogenic species *C. diphtheriae*, *C. ulcerans*, *C. pseudotuberculosis*, *C. rouxii*, *C. belfanti* and *C. salvaticum*. Samples were collected from a total of 18, 308 animals exhibiting various illnesses; 64 isolates belonged to the Cdc group and 26 of these harbored the tox gene. 51 of the cases were associated with *C. ulcerans*, while *C. diphtheriae* accounted for two cases and *C. rouxii* made up the remaining 11 cases. This is the largest

study of its kind ever performed with animals and provides interesting and useful information on the presence of pathogenic *Corynebacterium* species in human companion animals, which included dogs, cats, rabbits and horses. Since many of the infections associated with the Cdc group of bacteria were polymicrobial, it could not be determined with certainty that Cdc organisms were the cause of these infections. The Discussion section is well written and provides a detailed informative narrative on the zoonotic potential of *C. ulcerans* and *C. rouxii*.

Comments:

1. The authors provide extensive information on natural animal infections associated with bacteria from the Cdc group, but have not discussed or mentioned the results of any laboratory animal models using these organisms (primarily *C. ulcerans*). A discussion on this topic would be a useful addition to the Discussion section.

Reviewer #2 (Comments for the Author):

Major Comments

This is a remarkable study of the taxa closest to *C. diphtheriae* as recovered from a wide number of different infected animals which had been referred for examination in a veterinary setting in France.

Objectives/Introduction:

Use of the term 'diphtheriae complex' is a very useful phrase particularly as based on the comprehensive discussion in this paper, but it must be acknowledged for readers that this phrase is still not widely used in the literature. It is suggested that the authors more precisely define/defend which species are/should be referred to by this phrase, possibly incorporating some of well defined genetic lineages from recent publications. *C. diphtheriae*, *C. rouxii* and *C. belfantii* are closest to each other (by 16S, other single target genes, MALDI-TOF, other) but are discernable using those common methods from *C. ulcerans*/*C. pseudotuberculosis*/*C. silvaticum*. The ability/inability of any of these 6 species to produce diphtheria toxin has been well reviewed in this paper. However, possibly by using this term, clinicians/vets may opt not to identify or fail to discern among Cdc taxa to species level, such as identifying a strain of *C. diphtheriae* from *C. ulcerans*. Advanced genetic methods are definitive ways to discern among some of these species (*C. diphtheriae* from *C. rouxii*, for example). Therefore, it would be useful to state which species should be considered under the Cdc umbrella and clearly recommend that recovery of any of these species (by whatever means) should be forwarded to a reference lab for tox gene PCR followed by testing for expression of that toxin if tox gene positive (also described in the methods section).

The authors infer that the Bruker MALDI system can discern *C. rouxii* from *C. diphtheriae*. Authors should provide a reference which describes actual Biotyper scores vs identification done by genetic means (such as, *C. rouxii* 2.2, *C. diphtheriae* 1.4 etc) as this information was only inferred by the original *rouxii* publication. Authors should describe that *C. diphtheriae* and *C. belfantii* as well as that *C. ulcerans* from *C. silvaticum* can not be discerned by MALDI (being closely related by both 16S and rpoB sequencing differing by source etc).

Line 420, not all *C. ulcerans* strains derived from human disease were found to be 'inherently resistant' to clindamycin (in various older publications). The Marosevic papers 2020a, b (are these the same publication, entered 2x?) suggested that testing clindamycin using different methods and breakpoints (Eucast & CLSI) was problematic (result errors with respect to each other). The latest Eucast from January 2023 suggested that wild type *C. ulcerans* may be 'less susceptible' to clindamycin, not inherently resistant. Therefore this sentence should be modified and clearly state that clindamycin was not tested.

Minor Comments

Abstract methods, line 33, line 55, other places: add comma as in '18,308' or remove space.

Line 66: authors should modify wording to 'which may carry' the tox gene as a number of studies have shown that a large percentage of *C.d.* strains linked to infections lack this gene; this is also reviewed in this paper.

Line 141, correct spelling of isolates

Line 164, 167, more usual spelling is tetracyclines, not tetracylins

Line 236, correct spelling of conjunctivitis

Line 416, suggest removing 'as requested by law' as readers will not know what laws are implicated in this notation but presumably this is outlined in the website described.

Staff Comments:

Preparing Revision Guidelines

To submit your modified manuscript, log onto the eJP submission site at <https://spectrum.msubmit.net/cgi-bin/main.plex>. Go to

Author Tasks and click the appropriate manuscript title to begin the revision process. The information that you entered when you first submitted the paper will be displayed. Please update the information as necessary. Here are a few examples of required updates that authors must address:

Please return the manuscript within 60 days; if you cannot complete the modification within this time period, please contact me. If you do not wish to modify the manuscript and prefer to submit it to another journal, please notify me of your decision immediately so that the manuscript may be formally withdrawn from consideration by Microbiology Spectrum.

In this report, Museux et. al, present the results of case studies performed in France over a two year period that identified possible diphtheria infections in animals. The infections were associated with the presence of bacteria that are members of the Corynebacterium diphtheriae complex (Cdc), which includes the pathogenic species *C. diphtheriae*, *C. ulcerans*, *C. pseudotuberculosis*, *C. rouxii*, *C. belfanti* and *C. salvaticum*. Samples were collected from a total of 18, 308 animals exhibiting various illnesses; 64 isolates belonged to the Cdc group and 26 of these harbored the *tox* gene. 51 of the cases were associated with *C. ulcerans*, while *C. diphtheriae* accounted for two cases and *C. rouxii* made up the remaining 11 cases. This is the largest study of its kind ever performed with animals and provides interesting and useful information on the presence of pathogenic Corynebacterium species in human companion animals, which included; dogs, cats, rabbits and horses. Since many of the infections associated with the Cdc group of bacteria were polymicrobial, it could not be determined with certainty that Cdc organisms were the cause of these infections. The Discussion section is well written and provides a detailed informative narrative on the zoonotic potential of *C. ulcerans* and *C. rouxii*.

Comments:

1. The authors provide extensive information on natural animal infections associated with bacteria from the Cdc group, but have not discussed or mentioned the results of any laboratory animal models using these organisms (primarily *C. ulcerans*). A discussion on this topic would be a useful addition to the Discussion section.

Point-by-point answers to reviewers:

Reviewer comments:

Reviewer #1 (Comments for the Author):

In this report, Museux et. al, present the results of case studies performed in France over a two-year period that identified possible diphtheria infections in animals. The infections were associated with the presence of bacteria that are members of the *Corynebacterium diphtheriae* complex (Cdc), which includes the pathogenic species *C. diphtheriae*, *C. ulcerans*, *C. pseudotuberculosis*, *C. rouxii*, *C. belfanti* and *C. salvaticum*. Samples were collected from a total of 18, 308 animals exhibiting various illnesses; 64 isolates belonged to the Cdc group and 26 of these harbored the tox gene. 51 of the cases were associated with *C. ulcerans*, while *C. diphtheriae* accounted for two cases and *C. rouxii* made up the remaining 11 cases. This is the largest study of its kind ever performed with animals and provides interesting and useful information on the presence of pathogenic *Corynebacterium* species in human companion animals, which included dogs, cats, rabbits and horses. Since many of the infections associated with the Cdc group of bacteria were polymicrobial, it could not be determined with certainty that Cdc organisms were the cause of these infections. The Discussion section is well written and provides a detailed informative narrative on the zoonotic potential of *C. ulcerans* and *C. rouxii*.

Our answer: thank you for your positive comments.

Comments:

1. The authors provide extensive information on natural animal infections associated with bacteria from the Cdc group, but have not discussed or mentioned the results of any laboratory animal models using these organisms (primarily *C. ulcerans*). A discussion on this topic would be a useful addition to the Discussion section.

Our answer: thank you for the suggestion. The use of animal models is indeed worth mentioning along with our report. We have therefore added some words on this topic: "and experimental intranasal and intravenous infection studies in mice showed the pathogenic capacity of *C. ulcerans*, irrespective of toxigenicity (Dias et al., 2011; Mochizuki et al., 2016)."

Reviewer #2 (Comments for the Author):

Major Comments

This is a remarkable study of the taxa closest to *C. diphtheriae* as recovered from a wide number of different infected animals which had been referred for examination in a veterinary setting in France.

Our answer: thank you for your positive comments.

Objectives/Introduction:

Use of the term 'diphtheriae complex' is a very useful phrase particularly as based on the comprehensive discussion in this paper, but it must be acknowledged for readers that this phrase is

still not widely used in the literature. It is suggested that the authors more precisely define/defend which species are/should be referred to by this phrase, possibly incorporating some of well defined genetic lineages from recent publications. *C. diphtheriae*, *C. rouxii* and *C. belfantii* are closest to each other (by 16S, other single target genes, MALDI-TOF, other) but are discernable using those common methods from *C. ulcerans*/*C. pseudotuberculosis*/*C. silvaticum*. The ability/inability of any of these 6 species to produce diphtheria toxin has been well reviewed in this paper. However, possibly by using this term, clinicians/vets may opt not to identify or fail to discern among Cdc taxa to species level, such as identifying a strain of *C. diphtheriae* from *C. ulcerans*. Advanced genetic methods are definitive ways to discern among some of these species (*C. diphtheriae* from *C. rouxii*, for example). Therefore, it would be useful to state which species should be considered under the Cdc umbrella and clearly recommend that recovery of any of these species (by whatever means) should be forwarded to a reference lab for tox gene PCR followed by testing for expression of that toxin if tox gene positive (also described in the methods section).

Our answer: thank you for your suggestion on the definition of the *C. diphtheriae* species complex. We have modified the first introduction sentence as follows: “This bacterial species is phylogenetically related to 5 other *Corynebacterium* species (*C. ulcerans*, *C. pseudotuberculosis*, *C. rouxii*, *C. belfantii* and *C. silvaticum*) and together with these, is grouped into the *C. diphtheriae* species complex (CdSC).” Note that we now propose to use *C. diphtheriae* species complex, rather than simply *C. diphtheriae* complex, to make it more explicit that it is a taxonomic definition. We here follow a similar denomination in other pathogens (e.g., the *Klebsiella pneumoniae* species complex, abbreviated KpSC – see for example Margaret M C Lam, Ryan R Wick, Louise M Judd, Kathryn E Holt, Kelly L Wyres *Microb Genom* 2022 Mar;8(3):000800. doi: 10.1099/mgen.0.000800 Kaptive 2.0: updated capsule and lipopolysaccharide locus typing for the *Klebsiella pneumoniae* species complex). As for the recommendation to refer isolates of the CdSC to a reference lab, we have added the following sentence in the Discussion: “We suggest that in case of CdSC detection in animals, the veterinary laboratories should be encouraged to send the samples to a reference laboratory for analysis of the presence of the *tox* gene.”

The authors infer that the Bruker MALDI system can discern *C. rouxii* from *C. diphtheriae*. Authors should provide a reference which describes actual Biotyper scores vs identification done by genetic means (such as, *C. rouxii*, 2.2, *C. diphtheriae* 1.4 etc) as this information was only inferred by the original *rouxii* publication. Authors should describe that *C. diphtheriae* and *C. belfantii* as well as that *C. ulcerans* from *C. silvaticum* can not be discerned by MALDI (being closely related by both 16S and *rpoB* sequencing differing by source etc).

Our answer: thank you for your comment. In fact we do not state that MALDI differentiates *rouxii* from diphtheria. On the contrary, we state: “*C. rouxii* was identified retrospectively by MLST, as this species was only recognized as a separate species during the study period (in 2020), and as MALDI-TOF still identifies *C. rouxii* as *C. diphtheriae*, based on the MALDI Biotyper Compass database version 4.1 (version (100)”. We have clarified this in the methods: “Note that this method identifies *C. diphtheriae* and *C. ulcerans*, but does not discriminate well other members of the CdSC from these two species”.

Line 420, not all *C. ulcerans* strains derived from human disease were found to be 'inherently resistant' to clindamycin (in various older publications). The Marosevic papers 2020a, b (are these the same publication, entered 2x?) suggested that testing clindamycin using different methods and

breakpoints (Eucast & CLSI) was problematic (result errors with respect to each other). The latest Eucast from January 2023 suggested that wild type *C. ulcerans* may be 'less susceptible' to clindamycin, not inherently resistant. Therefore this sentence should be modified and clearly state that clindamycin was not tested.

Our answer: thank you. We have modified by replacing “inherently resistant” to “less susceptible”, and we already stated in the initial version: “Therefore, clindamycin was only tested for *C. diphtheriae* and *C. rouxii*”.

Minor Comments

Abstract methods, line 33, line 55, other places: add comma as in '18,308' or remove space.

Our answer: thank you, done.

Line 66: authors should modify wording to 'which may carry' the tox gene as a number of studies have shown that a large percentage of *C.d.* strains linked to infections lack this gene; this is also reviewed in this paper.

Our answer: thank you; we believe that “which carry” refers to the toxigenic strains, so the statement is correct.

Line 141, correct spelling of isolates

Our answer: thank you, done.

Line 164, 167, more usual spelling is tetracyclines, not tetracylins

Our answer: thank you, done.

Line 236, correct spelling of conjunctivitis

Our answer: thank you, done.

Line 416, suggest removing 'as requested by law' as readers will not know what laws are implicated in this notation but presumably this is outlined in the website described.

Our answer: thank you, done.

March 7, 2023

Dr. Sylvain Brisse
Institut Pasteur
Biodiversity and Epidemiology of Bacterial Pathogens
25-28 rue du Dr Roux
Paris 75724
France

Re: Spectrum00006-23R1 (Corynebacterium of the diphtheriae complex in companion animals: clinical and microbiological characterization of 64 cases from France)

Dear Dr. Sylvain Brisse:

Your manuscript has been accepted, and I am forwarding it to the ASM Journals Department for publication. You will be notified when your proofs are ready to be viewed.

Sincerely,

Florence Doucet-Populaire
Editor, Microbiology Spectrum
